# Mechanisms during Osteogenic Differentiation in Human Dental Follicle Cells

**DOI:** 10.3390/ijms23115945

**Published:** 2022-05-25

**Authors:** Christian Morsczeck

**Affiliations:** Department of Oral and Maxillofacial Surgery, University Hospital Regensburg, Franz-Josef-Strauss-Allee 11, 93053 Regensburg, Germany; christian.morsczeck@klinik.uni-regensburg.de; Tel.: +49-(0941)-944-6310; Fax: +49-(0941)-944-6372

**Keywords:** dental follicle cells, osteogenic differentiation, signaling pathways, tooth development, tooth eruption

## Abstract

Human dental follicle cells (DFCs) as periodontal progenitor cells are used for studies and research in regenerative medicine and not only in dentistry. Even if innovative regenerative therapies in medicine are often considered the main research area for dental stem cells, these cells are also very useful in basic research and here, for example, for the elucidation of molecular processes in the differentiation into mineralizing cells. This article summarizes the molecular mechanisms driving osteogenic differentiation of DFCs. The positive feedback loop of bone morphogenetic protein (BMP) 2 and homeobox protein DLX3 and a signaling pathway associated with protein kinase B (AKT) and protein kinase C (PKC) are presented and further insights related to other signaling pathways such as the WNT signaling pathway are explained. Subsequently, some works are presented that have investigated epigenetic modifications and non-coding ncRNAs and their connection with the osteogenic differentiation of DFCs. In addition, studies are presented that have shown the influence of extracellular matrix molecules or fundamental biological processes such as cellular senescence on osteogenic differentiation. The putative role of factors associated with inflammatory processes, such as interleukin 8, in osteogenic differentiation is also briefly discussed. This article summarizes the most important insights into the mechanisms of osteogenic differentiation in DFCs and is intended to be a small help in the direction of new research projects in this area.

## 1. Introduction

Molecular processes during osteogenic differentiation of dental follicle cells (DFCs) have been studied for many years. However, to better understand the greater importance of this very specific subject, a brief look at the role of the tooth in the formation of mineralized tissue in animals is helpful.

Teeth probably comprise evolutionarily the earliest mineralized tissue, having been found in the feeding apparatus of extinct jawless fish, the conodonts [1]. Nonetheless, the evolution of the process of biomineralization in jawed vertebrates is a complex process that can be traced even in extant animals. The elasmobranch group, such as whale sharks, can produce calcified cartilage and skin bone, including teeth, denticles, and fin spines, but do not make endochondral mineralization because they lack genes from the secretory calcium-binding phosphoprotein (SCPP) gene family [2]. Interestingly, dental mineralization like most craniofacial mineralized tissues also does not undergo the process of endochondral mineralization. This fact is very important as it suggests that tooth development and, in particular the processes of biomineralization, are different from skeletal bone development. This makes mineralizing dental cells very special for analyses of molecular processes during osteogenic differentiation of mammalian tissues. The dental follicle is part of the tooth germ and its cells are the true progenitors of two mineralizing cells: cementoblasts and alveolar osteoblasts [3,4]. However, before going into detail about dental follicle cells, some short general remarks about the dental follicle in tooth development should be made.

The tooth germ can initially be divided into two different cell types based on the origin of their germ layers. One cell type is derived from the ectodermal germ layer (the origin of the enamel-forming ameloblasts) and the other—dental mesoderm—is derived from neural crest cells. Tooth development generally represents an interaction between these ectodermal and mesodermal cell layers that form the three dental germ tissues: enamel organ, dental papilla (dental pulp progenitor) and dental follicle [5,6]. During the early stages of tooth development, the dental follicle, sometimes called the “dental sac“, forms a sac-like tissue around the other two tooth germ tissues (Figure 1). The dental follicle takes part in the morphology of the tooth crown and it is also involved in the developmental processes of tooth root formation [7,8]. Cells from the dental follicle (DFCs) are therefore also precursors of all mineralizing and non-mineralizing cells of the periodontium.

The dental follicle, but not the developing tooth crown, actively participates in tooth eruption as it allows the eruption of an encapsulated metal crown replacing a tooth crown at an early stage of tooth development [9]. While the exact role remains elusive, several tooth eruption factors have been discovered in the dental follicle over time. These include parathyroid hormone-related protein (PTHrP), epidermal growth factor (EGF), interleukin (IL)-1α, colony-stimulating factor-1 (CSF-1), monocyte chemotactic protein-1 (MCP-1) and their respective receptors, proto-oncogenes, and transcription factors such as c-Fos, nuclear factor-kappa B 1 (NFκB1) [10,11,12]. Two of these factors, PTHrP and NFκB1, which are induced by RANK/RANKL (Receptor activator of nuclear factor kappa-Β/-ligand), are also involved in the osteogenic differentiation of dental follicle cells and their functions in this context are discussed in more detail below. Moreover, these proteins and their pathways are required for the maturation and migration of alveolar-osteoclasts that allow tooth eruption. A major step for osteoclastogenesis and for bone resorption is the reduced expression of the osteoclastogenesis inhibitory factor osteoprotegerin (OPG) in the dental follicle, which is mediated by the expression of colony-stimulating factor-1 (CSF-1) and parathyroid hormone-related protein (PTHrP) in the tooth germ [10,13]. In contrast, for alveolar bone formation, OPG secretion of DFCs is enhanced and can be further enhanced by the application of osteogenic differentiation inducer bone morphogenetic protein-2 (BMP-2) [14]. These studies support the hypothesis that the dental follicle coordinates antagonistic processes involving the regulation/activation of immune cells, particularly osteoclasts, for tooth eruption and directing the differentiation of undifferentiated cells within the dental follicle into functional tissue cells of the periodontium [15]. The isolation and detailed characterization of DFCs was therefore an important step in the understanding of both tooth eruption and the development of the periodontium.

This article can be subdivided. First, it briefly discusses the isolation of human dental follicle cells. Then, it describes molecular mechanisms driving osteogenic differentiation of DFCs and ends with a short conclusion.

## 2. Isolation of Cells from Human Dental Follicles

For the beginning of the endeavor of the isolation of undifferentiated cells from the dental follicle two properties were actually suggested: these cells have the potential to differentiate into functional tissue cells such as alveolar osteoblasts and they have the ability to self-renew (to become a new undifferentiated cell again). These undifferentiated cells like other somatic stem cells have the ability to maintain their undifferentiated state even after numerous cell division cycles. They can renew themselves with symmetrical as well as asymmetrical cell division so that they are potentially immortal without having the typical characteristics of tumor cells [16,17,18,19]. Interestingly, tumor biology has shown, for example, that somatic stem cells of oral squamous tissue cells can become tumor-initiating cells, with the number of stem cells in tissue frequently increasing [20,21]. These stem cells are also known as tumor stem cells. The number of somatic stem cells in tissues can increase when they divide symmetrically. However, it is assumed that the symmetrical cell division of somatic stem cells is rather rare since the number of stem cells in adult tissue remains constant [16]. Here, after the division of the stem cell, another stem cell is created, but also another cell that is no longer a stem cell, but in which the differentiation process has already been induced. This cell has only a very limited ability to renew itself, which was shown, for example, in the dental epithelium [17,22]. After the differentiation into a functional body cell was initiated, it depends on the origin (tissue/organ) of the stem cell into which cell type it can differentiate. It should be noted here that differentiation of dental stem cells into tissue cells other than those of their own tissue—for example in neural cells—is possible under defined in vitro conditions [23,24]. It is assumed that the differentiation or self-renewal of stem cells depends on various factors that influence their behavior. These factors are: 1. soluble factors, such as growth factors, but also 2. cell–cell contacts, 3. cell–extracellular matrix contacts and 4. mechanical properties of the surrounding tissue. These factors define among others the so-called stem cell niche, which is not only responsible for controlling self-renewal but also regulates the targeted induction of differentiation into functional tissue cells [17].

While these factors remained elusive for protocols for the isolation of undifferentiated cells, the first isolation of DFCs adopted a simple method used decades ago by osteologist Friedenstein to isolate bone marrow stromal cells (BMSCs) [25]. Here, after tissue isolation and preparation of single cells, cells were selected that were able to adhere to plastic and form colonies [26]. Since wisdom teeth are frequently extracted for dental reasons, various tissues can be preserved from the tooth germ for the isolation of undifferentiated cells. These tissues of impacted wisdom teeth are the dental follicle, which can be divided into coronal and periapical follicles (Figure 1), and the apical dental papilla, as well as the dental pulp and periodontium of matured and fully erupted wisdom teeth [27]. The undifferentiated cells of the dental follicle (DFCs) were actually isolated through their plastic adherence as colony-forming cells in a serum-containing cell culture medium [25]. These fibroblast-like cells express typical stem cell markers and have good osteogenic differentiation potential, which can be used to analyze the molecular processes of osteogenic differentiation from dental stem cells to alveolar osteoblasts [28].

However, additional cell types were isolated from the dental follicle [27]. Recently, Oh and Yi isolated dental follicle-derived Hertwig’s epithelial root sheath cells with a keratinocyte serum-free cell culture medium [29]. These cells showed typical features of epithelial cells unless epithelial–mesenchymal transition (EMT) was induced by treatment with fetal bovine serum, which is part of the DFC standard medium. Afterward, EMT-isolated epithelial cells showed typical features of mesenchymal cells, for example, they differentiated into mineralizing cells. One can therefore assume that the origin of DFCs, which can be isolated as colony-forming cells by plastic attachment, may also have been epithelial cells of the dental follicle after EMT [29]. However, it remains unclear to what extent these cells can be distinguished from DFC derived from the dental mesoderm.

Interestingly, moreover, subpopulations of murine dental follicle cells were discovered in murine tooth germs. Ono and colleagues showed that murine dental follicles contain different subpopulations that can be distinguished by expressing specific genes such as the osteogenic transcription factor Osterix, PTHrP, and Gli1, which are involved in the Hedgehog signaling pathway, which contributes to both tooth eruption and periodontal development [4,8,30]. However, how these murine DFC populations relate to isolated human DFCs is unknown and is a worthwhile research topic for the future. These plastic adherent, fibroblast-like human DFCs were widely used for studies analyzing the molecular processes of differentiation of DFCs into cementoblasts or alveolar osteoblasts, and the following part of this article focuses on studies on these molecular mechanisms.

## 3. Molecular Mechanisms of the Osteogenic Differentiation of DFCs

Isolated undifferentiated stem/progenitor cells are particularly suitable, among other things, for studying developmental processes under in vitro conditions. Molecular processes during osteogenic differentiation are only one example. Molecular studies with different osteogenic progenitor cells revealed which pathways are involved in the induction and in the regulation of the osteogenic differentiation, which is difficult to obtain with primary alveolar osteoblasts or with immortalized tumor cell lines. Here, the integration of the Bone Morphogenetic Protein (BMP) signaling pathway, the canonical WNT (Wingless-Type MMTV Integration Site Family Member) signaling pathway, the Mitogen-Activated Protein Kinase (MAPK) signaling pathway and the NOTCH signaling pathway is crucial for the osteogenic differentiation process [31,32,33]. Moreover, additional signaling pathways and transcription factors in DFCs during osteogenic differentiation were discovered and analyzed, which are discussed in the following sections.

This part of the article begins with the positive feedback loop of bone morphogenetic protein (BMP)2 and homeobox protein DLX3 and continues with a signaling pathway associated with protein kinase B (AKT) and protein kinase C (PKC). These two signaling pathways are presented in a larger context. Then short summaries on the role of non-coding RNAs and epigenetic modifications during differentiation, followed by first insights into the influence of extracellular matrix proteins and biological processes, such as, e.g., B. cellular senescence, on the induction of osteogenic differentiation. Thereafter, work examining the role of pro- and anti-inflammatory factors in osteogenic differentiation is discussed.

### 3.1. The BMP2/DLX3 Positive Feedback Loop

Downstream and sometimes also upstream of the above-mentioned signaling pathways, specific transcription factors are involved in the differentiation of osteogenic progenitor cells [31]. Interestingly, while transcription factors such as runt-related transcription factor (RUNX)2, which is generally involved in the process of osteogenic differentiation, are constitutively expressed in DFCs during osteogenic differentiation, only the homeobox gene DLX3 was differentially induced [34]. DLX3 belongs to the “Distal-less (Dlx)” family with a homeobox protein domain. This protein is essential for tooth and bone development [35,36,37]. A mutation of the DLX3 gene, for example, is the cause of the autosomal dominant tricho dento osseous (TDO) syndrome, which is associated with increased bone mineral density and thickness in the craniofacial bones [38,39,40,41]. It was shown that DLX3, along with other transcription factors such as DLX5, play an important role in BGLAP gene (osteocalcin) transcriptional activity and in the induction of BMP2-mediated RUNX2 expression [35]. Moreover, DLX3 directly affected the cell viability of DFCs and supports the osteogenic differentiation [34,42].

While comparisons of gene and protein expression profiles between DFCs before and after the osteogenic differentiation revealed that only a small percentage of the regulated genes are associated with the biological process of skeletal development [43], the BMP signaling pathway is associated with both the osteogenic differentiation process and DLX3 expression [42,44]. The BMP signaling pathway belongs to the family of the TGF-β signaling pathway, which is down-regulated in DFC sub-clones with reduced osteogenic differentiation potentials [45]. However, the impact of the TGF-β signaling pathway on the differentiation potential is complex because supplementation of the osteogenic differentiation medium with TGF-β inhibits the osteogenic differentiation of DFCs [46]. In contrast, BMPs are well-known osteogenic growth factors and in particular, BMP2, BMP7 and BMP9 offer high potential for stimulating osteogenic differentiation or cementogenic differentiation in DFCs [47,48]. The binding of BMP growth factors to BMP type 1 and type 2 receptors, which is a serine/threonine receptor kinase, induces BMP canonical signaling. It phosphorylates proteins such as SMAD1/5/8, which migrate to the cell nucleus and act as a transcription factor to induce gene expression. Interestingly, after induction of osteogenic differentiation with BMP2, DLX3 is not only up-regulated but in turn induces the BMP signaling pathway [42]. Thus, a BMP2/DLX3 positive feedback loop supports osteogenic differentiation to a point, since further induction of BMP2 or DLX3 expression did not further increase the osteogenic differentiation and the expression of osteoblast markers such as alkaline phosphatase activity [49]. In order to understand this regulation, additional signaling pathways that interact with the BMP2/DLX3 signaling pathway were also examined.

Analyses of phosphoproteomes of DFCs after induction of osteogenic differentiation revealed that BMP2 induced hedgehog “off” state phosphoproteins [50]. Furthermore, BMP2 induced the expression of hedgehog signaling repressors such as Patched 1 (PTCH1), Suppressor of Fused (SUFU) and PTHrP, thereby inhibiting hedgehog signaling after induction of osteogenic differentiation [50]. However, this result suggests that the role of PTHrP after secretion into the extracellular environment is complex and cannot, therefore, relate solely to its function in tooth eruption [51](see above). The secreted protein PTHrP is required for a variety of different processes including regulation of endochondral bone development, differentiation of bone progenitor cells and the development of craniofacial tissue [4,52]. During endochondral osteogenic differentiation, for example, the Indian Hedgehog (IHH) induces PTHrP and in turn, PTHrP inhibits IHH via a negative feedback loop. The PTHrP protein is also secreted after induction of osteogenic differentiation of DFCs. However, neither hedgehog signaling induced PTHrP nor did PTHrP repress hedgehog signaling during osteogenic differentiation in DFCs [53]. While supplementation of the osteogenic differentiation medium with PTHrP inhibited the expression of DLX3, inhibition of PTHrP secretion did not support the osteogenic differentiation of DFCs. Thus, secreted PTHrP regulates osteogenic differentiation in DFCs independently of hedgehog signaling [53]. Interestingly, however, PTHrP has additional molecular functions. In a recent study with DFCs, a particular cell line had a high endogenous expression of PTHrP, which is located in the cell nucleus and not secreted into the extracellular space [54]. Only parts of the PTHrP protein move into the cell nucleus and influence protein functions, for example, cell cycle proteins [55]. This DFC cell line demonstrated an improved induction of osteogenic differentiation with high alkaline phosphatase (ALP) activity and reliable mineralization potential. While known osteogenic markers such as RUNX2 were similarly expressed compared to other cell lines with reduced PTHrP expression, bone morphogenetic protein (BMP) was also more highly expressed. In this cell line, the inhibition of PTHrP gene expression by specific siRNAs reduces ALP activity and BMP signaling. These effects were submitted by nuclear-localized PTHrP because an inhibitor for the PTHrP receptor did not affect osteogenic differentiation [54]. Thus, PTHrP may have antagonistic functions in the osteogenic differentiation of DFCs. The secreted protein regulates while the nuclear-localized protein supports osteogenic differentiation.

In addition, the NOTCH signaling pathway also regulates the osteogenic differentiation of DFCs [56]. The family of NOTCH proteins are transmembrane receptors that mediate communication between neighboring cells and are involved in a number of developmental processes [57]. The binding of the NOTCH ligand triggers the cleavage of an intracellular domain of NOTCH (NICD) translocates into the nucleus and affects the transcription like a transcription factor. One member of the family is NOTCH-1 which is not only a marker of human DFCs but also induced during osteogenic differentiation by the BMP2/DLX3 pathway. This increased expression of NOTCH-1 in DFCs not only inhibited the BMP2/DLX3 positive feedback loop but increased cell proliferation and self-renewing [56].

Furthermore, the osteogenic differentiation of DFCs involving the BMP2/DLX3 pathway also depends on parts of the canonical WNT/β-catenin signaling pathway [49]. The β-catenin protein is the key player in the WNT signaling pathway and is strongly induced upon activation of the WNT receptors, for example, by WNT3A. This activated β-catenin protein translocates to the nucleus and stimulates lymphoid enhancer-binding factor/T cell factor (LEF/TCF1) DNA binding and the expression of specific target genes. Confusingly, while the expression of β-catenin is important for DLX3 expression and osteogenic differentiation [49], the induction of the WNT/β-catenin signaling by WNT3A not only inhibits osteogenic differentiation of DFCs but also the gene expression of DLX3 [49]. In addition, depletion of adenomatosis polyposis coli down-regulated (APCDD) 1, a known inhibitor of canonical WNT signaling, inhibits not only β-catenin activation but also osteogenic differentiation of DFCs [58]. Interestingly, similar results were obtained with the canonical WNT signaling pathway inhibitor Naked Cuticle 2 (NKD2), which promotes both osteogenic differentiation and β-catenin expression in rat DFCs [59]. In addition, DKK1, another inhibitor, reduces the inhibitory effects of tumor necrosis factor (TNF)-α on osteogenic differentiation of DFCs [60]. However, while supplementation of osteogenic differentiation medium with DKK1 supported the osteogenic effect in human DFCs [49], it inhibited the differentiation of rat DFCs [60]. These results suggest that inhibitory molecules of the canonical WNT pathway support both β-catenin and—at least under certain circumstances—the osteogenic differentiation in DFCs. Interestingly, β-catenin is also activated by protein kinase A (PKA), which itself is induced by BMP2 [49,61]. Activation of PKA supports osteogenic differentiation not only via activation of β-catenin and downstream induction of DLX3 expression but also through “off state” of the hedgehog signaling pathway, which can control osteogenic differentiation of DFCs [50]. While activation of PKA/β-catenin by BMP2 is still not fully understood, recent work suggests that PKA in DFCs is activated via nuclear-localized PTHrP, which is also induced by BMP2. In contrast, PKA inhibits BMP2 signaling and expression of PTHrP in a negative feedback loop and controls the BMP2/DLX3 pathway [61].

In summary, the BMP2/DLX3 positive feedback loop is involved in the osteogenic differentiation of DFCs and is controlled by PTHrP, PKA, and β-catenin, among others. Figure 2 summarizes this complex pathway.

### 3.2. Protein Kinases C (PKC) and B (AKT)

In addition to the BMP2/DLX3 pathway described above, there are other signaling pathways that are involved in differentiation in conjunction with it, but also independently of it. The transcription factor EGR1, which is involved in many biological processes, supports craniofacial development and osteogenic differentiation of DFCs [62,63]. Reumann et al. for example showed that depletion of EGR1 in mice affects the structure of cortical bones and might be crucial for general bone characteristics [64]. In DFCs the induction of EGR1 expression is submitted by the protein kinase-based pathway AKT also known as protein kinase B (PKB), which is induced during osteogenic differentiation with BMP2 [65]. Interestingly, EGR1 binds not only to the phosphorylated form of SMAD1/5 but it induces the osteogenic differentiation and the expression of BMP2 and DLX3 [63,65]. These results suggest that the AKT signaling pathway enables positive feedback between BMP2 and transcription factors EGR1 and DLX3. However, the role of AKT is more complex than just indicated. AKT was induced at an early phase (3 days) of the osteogenic differentiation with BMP2 in one specific cell line [65], but was decreased in another DFC cell line [66]. In this later cell line, the activation of AKT also disturbed the BMP-signaling pathway and the mineralization [66]. Thus, AKT and BMP signaling pathways communicate with each other, but in different DFC cell lines, the influence seems to be the opposite. The impact of the AKT pathway on mineralization is also ambiguous, as inhibition of AKT supports BMP2-induced mineralization, while both up- and down-regulation of AKT disrupt mineralization with dexamethasone [66]. Its relationship to classical protein kinase C (PKC) kinases such as PKCα adds complexity to the role of AKT in osteogenic differentiation. The activity of PKCα is sustained by PTHrP but inhibited by WNT5A, supporting the viability and osteogenic differentiation of DFCs [66,67]. Therefore, classical PKCs inhibit the osteogenic differentiation of various DFC cell lines, although they do not interfere with the induction of differentiation [66]. However, inhibition of PKCs induces the activity of AKT several days after induction of differentiation, supporting activation of osteogenesis and nuclear localization of β-catenin [66,68]. Moreover, the nuclear factor “kappa-light-chain-enhancer” of the activated B cells (NF-κB) pathway, which disturbs the mineralization of differentiated DFCs, is induced by classical PKCs but inhibited via the AKT axis [66]. However, the activation of the NF-κB pathway is not the sole cause for the inhibitory effect of classical PKCs for the osteogenic differentiation because inhibition of NF-κB could not fully restore the mineralization of DFCs after PKC activation [66]. Interestingly, a recent study suggests that a PKC-dependent pathway contributes to a cell-free DFC-derived matrix vesicle-mediated approach to alveolar bone regeneration [69], suggesting an additional role for PKC in the differentiation of alveolar bone progenitor cells. However, Figure 3 summarizes the PKC/AKT pathway during the osteogenic differentiation of DFCs.

New results also suggest that a complex of signaling pathways and crucial cellular processes directs the osteogenic differentiation. This was for example indicated by the sensitivity of osteogenic differentiation to changes in AMP-activated protein kinase (AMPK) and autophagic activity [70,71]. Both AMPK and autophagy antagonize the AKT pathway and are involved in the regulation of crucial and sometimes exclusive metabolic processes such as fatty acid biosynthesis or glycolysis involved in stress survival, cell growth and proliferation [72].

### 3.3. Epigenetics and Non-Coding RNAs

While signaling pathways guide the progress of osteogenic differentiation, this network of signaling pathways must be controlled not only by interactions such as feedback loops but also by additional cellular mechanisms. Therefore, the expression of osteogenic differentiation markers and also of signaling pathways can also be modulated by non-coding (nc) RNAs [73]. One group of ncRNAs are small RNAs (about 22 nucleotides), which can be obtained from ds RNAs or from RNA molecules with a hairloop structure. The group of small RNAs can be divided into different sub-groups with microRNAs (miR), which silence gene expression post-transcriptionally, as one prominent member. Some miRNAs are known to be involved in the differentiation of dental stem cells [73]. An example is miR-101, which not only actively regulates the expression of PLAP1, a cell marker of the periodontal ligament [74], but also induces and supports the osteogenic differentiation of DFCs [75]. Moreover, miR-101 appears to be a general inducer of osteogenic differentiation, as it also promotes osteogenic differentiation of adipose-derived mesenchymal stem cells [76]. Interestingly, a current study demonstrated that stem cell-derived exosomes contain miR101 which augments the differentiation in osteogenic progenitor cells [77]. While little is known about molecular targets in DFCs, miR-101 is involved in the expression of DLX3 and the activity of ALP [75]. In contrast to miR-101, miR-204 is down-regulated during osteogenic differentiation of DFCs and binds to the three prime untranslated regions (3′-UTR) of ALP and RUNX2 messenger RNAs [78]. Thus, miR-204 inhibits both expression of RUNX2 and the activity of ALP and is, therefore, an inhibitor of osteogenic differentiation in DFCs.

The second group of ncRNAs is the so-called long non-coding (lnc) RNAs. These nucleic acid molecules are at least 200 nt in size and they have diverse molecular functions during osteogenic differentiation, including regulation of gene expression at the transcriptional and post-transcriptional levels. Recently, a number of lncRNAs were discovered that promote or inhibit the differentiation of osteogenic progenitor cells [73,79]. One example of an inhibitory lncRNA is MEG3 (maternally expressed 3), which is down-regulated in DFCs compared to expression in PDL stem cells (PDLSCs) [80]. Furthermore, lncMEG3 is also down-regulated during osteogenic differentiation as it promotes EZH2 (Enhancer of zeste homolog 2), a histone methyltransferase component of the Polycomb Repressive Complex 2 (PRC2), which binds target genes, which then exhibit increased levels of inhibitory methylated histone proteins (H3K27me3). These targeted genes include WNT signaling pathway genes [80]. However, lncMEG3 supports the expression of known pluripotency markers such as NANOG or OCT-4 in undifferentiated DFCs, which also inhibits the induction of differentiation [81]. The second example of lncRNAs and osteogenic differentiation is HOTAIRM1 (HOXA transcript antisense RNA myeloid 1), which is induced during osteogenic differentiation. [82]. It is an antisense transcript known to regulate HOXA genes differentially expressed during myelopoiesis and neurogenesis [83]. However, Che et al. showed that HOXA2 and lncRNA HOTAIRM1 are similarly expressed in DFCs and that both gene products support osteogenic differentiation. They also demonstrated that repression of HOXA2 gene expression is achieved by DNA methylation of two CpG islands (CpG1 and CpG2) of the HOXA2 promoter region with DNA (cytosine 5) methyltransferase 1 (DNMT1) [82]. However, lncRNA HOTAIRM1 prevented the binding of DNMT1 to the HOXA2 promoter and the repression by DNA methylation.

These examples with ncRNAs already indicate that epigenetic modifications of both genomic DNA and histone proteins are involved in osteogenic differentiation. Besides the ncRNA, the ubiquitin-like SUMO system is another factor involved in the methylation of nuclear histone proteins and the regulation of gene expression. The SUMO-specific isopeptidase SENP3 controls the methylation of the histone protein H3K4 by regulating the histone-modifying SET1/MLL complex, which consists of histone methyltransferase and the regulatory components WDR5, RbBP5, Ash2L and DPY-30. MLL1/MLL2 complexes are particularly important for the activation of homeobox genes such as DLX3 [84]. Moreover, SENP3 associated with MLL1/MLL2 complexes catalyzes deSUMOylation of the regulatory component RbBP5, which is required for activation of gene transcription of DLX3. The absence of SENP3 reduced H3K4 methylation of histone proteins at the DLX3 promoter and thereby reduced recruitment of active RNA polymerase II, which inhibits DLX3 gene transcription and osteogenic differentiation of DFCs. This describes the importance of balanced SUMOylation for the epigenetic control of the BMP2/DLX3 feedback loop [84]. While SENP3 is involved in the regulation of DLX3 gene expression also an upstream regulator of SENP3 was identified [85]. Here, flightless-I homolog (FLII), a member of the gelsolin family of actin-remodeling proteins, is a novel regulator of SENP3. FLII is associated with SENP3 and the MLL1/2 complex and it determines SENP3 recruitment and MLL1/2 complex assembly on the DLX3 gene [85]. Consequently, FLII is essential for the function of SENP3 during osteogenic differentiation of DFCs.

Recent studies also identified factors directly affecting epigenetics and osteogenic differentiation of DFCs. Chromodomain-helicase-DNA-binding protein 7 (CHD7), for example, recognizes epigenetic modifications such as histone protein methylation and regulates the assembly and organization of nucleosomes and promotes the expression of genes such as the receptor of PTHrP (PTH1R) [86]. Another protein associated with the initiation of gene expression is the AFF4 transcription elongation complex protein, which is a part of the super-elongation complex [87]. This super-elongation complex promotes gene transcription of typical osteogenic differentiation markers such as BGLAP (osteocalcin), DLX5, SP7 (osterix) and RUNX2. Moreover, it promotes ALP activity after induction of osteogenic differentiation in DFCs. AFF4 is also associated with epigenetic regulation of gene expression since AFF4 binds to the promoter of ALKBH1 a known demethylase and a critical regulator of epigenetics. Xiao et al. showed that the depletion of AFF4 can be supplemented by overexpression of ALKBH1 [87]. In summary, although little is known at all, these examples indicate that ncRNAs and epigenetics play a crucial role in the osteogenic differentiation process in DFCs.

### 3.4. Extracellular Matrix (ECM)

During tooth development, DFCs come into contact with various tissues and the various extracellular matrix (ECM) proteins, polysaccharides, glycoproteins and proteoglycans that are important for cell behavior. A modification of the surface of a cell culture dish with ECM proteins or polysaccharides such as agarose for example influences whether DFCs are cultivated as adherent cells or as spheroids in a 3D cell culture system [24,88]. Tissues have also different physical properties and DFCs must therefore cope with different surface stiffnesses, for example, the rigid stiffness of alveolar bone or dental cementum and the soft stiffness of the PDL. Stem or progenitor cells such as DFCs react to these mechanical properties [89,90] and of course to the properties of different ECM proteins such as collagen I or laminin [91,92,93].

ECM proteins activate intracellular signaling pathways such as the Extracellular-signal Regulated Kinases (ERK) pathway or the Focal Adhesion Kinase (FAK) pathway for the induction of the osteogenic differentiation [92,94,95]. Laminins are one example of ECM proteins, which are involved in the osteogenic differentiation of DFCs. This large family of heterotrimeric multidomain proteins has growth factor-like domains that can bind to known receptors for ECM proteins Integrins on cell membranes [96]. The binding of the ECM laminin to the integrin-α3/-β3 receptor, for example, supports differentiation in bone marrow-derived osteogenic precursor cells [97]. DFCs differentially express extracellular matrix proteins of the laminin group during osteogenic differentiation [98]. While the activity of the early osteogenic differentiation marker ALP was inhibited, laminin strongly stimulated mineralization, which is a marker for the late phase of osteogenic differentiation [93]. To do this, laminins bind to α2/-β1 integrins on the cell membrane and activate the FAK/ERK signaling pathway downstream.

The same intracellular signaling pathway is activated in DFCs by collagen type I, which makes up most of the mineralized tissues [92]. However, the ERK signaling cascade is generally induced during osteogenic differentiation of human DFCs [43], but can also be activated by collagen I independently of FAK. While activation of FAK is mandatory for induction of the early osteogenic marker ALP, activation of ERK is required for the expression of late markers such as OPN [92]. Recent publications on tooth development have shown that collagen I and its receptor discoidin domain receptor 2 (DDR2) are involved in tooth development and craniofacial bone regeneration. Here, Franceschi and co-workers showed that knocking out the DDR2 gene in murine calvarial osteoblasts, dental pulp cells, or PDL cells inhibited mineralization and the expression of osteogenic differentiation markers [99,100]. However, it is not very likely that similar results will be obtained with human DFCs, as collagen had not affected the mineralization of human DFCs in a previous study [92].

A decellularized mineralized dentin matrix can also be obtained from extracted teeth and this dentin matrix was successfully used together with DFCs for tooth root regeneration and differentiation induction experiments [101,102,103]. This dentin matrix induced both osteogenic and odontogenic markers such as collagen I, osteopontin, dentin sialophosphoprotein (DSPP) and dentin matrix protein (DMP)1. However, under in vivo conditions a combination of allogenic dentin-matrix and DFCs induced typical markers for inflammatory (T-helper cell 1) response (induction of interleukin-1 beta, TNF-α), although the formation of typical tooth root tissues was demonstrated [103]. In addition, DFCs did not appear to show any immunomodulatory properties in this study. Interesting results were also obtained by Yang et al. [102], they showed that a combination of dentin matrix and DFCs improved the regeneration in the periodontitis model, but the cell survival rate of DFCs was very low [104]. Interestingly, dentin induced oxidative stress in DFCs, which inhibited the osteogenic differentiation potential [104]. However, osteogenic differentiation potential was increased after oxidative stress was alleviated by N-acetylcysteine, a precursor of glutathione. These data suggest that cell viability and related biological processes such as senescence are of great importance for the osteogenic differentiation potential of DFCs.

### 3.5. Influence of Cell Viability and Cellular Senescence

As mentioned above, recent data have led us to believe that biological processes related to cell viability should receive a new focus for studies of osteogenic differentiation mechanisms in DFC. A process strongly associated with cell viability is cellular senescence. It is also known that differentiation in osteogenic progenitor cells is impaired after induction of cellular senescence [105,106,107,108,109]. DFCs acquire senescence at later stages of cell culture [110]. Here, decreased cell proliferation is associated with induction of β-Galactosidase activity and increased cell size, which is typical for senescent cells. Interestingly, while preliminary data showed that dental stem cell markers such as CD105 did not change significantly after induction of cellular senescence, the differentiation potential in senescent DFCs decreased [110,111]. In general, senescent cells control the progression of the cell cycle in the G1 phase by inhibition of cyclin-dependent kinases (CDKs). This can be carried out, for example, via the induced expression of the p21 protein, which is activated by the p53 (TP53) protein, or via the p16 protein expression. Both proteins are cell cycle regulator proteins and are indirectly or directly induced by oxidative stress, telomerase inhibition, oncogenes and enzymes that are involved in DNA repair [109,112,113]. Recent experiments showed that p16, but less p21, induced senescence in DFCs, but p16 inhibition did not restore the osteogenic differentiation in senescent DFCs [114]. However, the WNT5A inducer of the non-canonical WNT pathway is involved in both osteogenic differentiation and induction of cellular senescence [111,115]. Previous studies suggest that WNT5A supports the BMP2/DLX3-induced osteogenic differentiation and regulates the activation of β-catenin in DFCs [116,117]. My group could not confirm this, but it is very likely that WNT5A plays a role in cell viability and inhibition of induction of cellular senescence since it is gradually down-regulated after cellular senescence was induced; much like the expression of osteogenic differentiation markers in DFCs during induction of senescence [67]. Interestingly, WNT5A has an impact on osteogenic differentiation of senescent DFCs, but no significant impact on the expression of osteogenic differentiation markers prior to induction of senescence. This means that WNT5A is not directly involved in signaling pathways responsible for osteogenic differentiation, but supports the viability of senescent cells that are likely to be damaged [67]. Furthermore, TP53, which is an inducer of both cellular senescence and apoptosis, is also implicated in reduced osteogenic differentiation potential of senescent DFCs. In contrast to WNT5A, TP53 inhibits osteogenic differentiation in senescent DFCs but is not involved in the induction of senescence [118].

Recent studies suggest that mechanisms involved in the induction of cellular senescence are also related to the inhibition of osteogenic differentiation. Kampron et al. showed for example that uremic toxins not only inhibit osteogenic differentiation but also induce senescence [106]. It is very likely that these mechanisms are reversible. A recent study with bone marrow-derived mesenchymal stem cells showed that melatonin stimulated the expression of histone methyltransferase nuclear receptor binding SET domain protein 2 (NSD2) through MT1/2-mediated signaling pathways, resulting in the rebalancing of H3K36me2 and H3K27me3 modifications to increase chromatin accessibility of the osteogenic genes, runt-related transcription factor 2 (RUNX2) and bone gamma-carboxyglutamate protein (BGLAP, osteocalcin) [119]. This study demonstrates that inhibitory processes of osteogenic differentiation associated with epigenetic changes during cellular senescence are reversible. However, additional biological processes are also related to the cellular senescence of DFCs. Bastos et al. showed that metabolic processes are highly regulated in senescent dental follicles [12]. This result correlates very well with the role played by the osteogenic transcription factor ZBTB16 in the osteogenic differentiation of DFCs and in crucial metabolic processes [120,121,122]. Unfortunately, metabolic processes in DFCs are poorly understood. Basic metabolic processes could be key not only to understanding molecular mechanisms during osteogenic differentiation but also to renewal differentiation potential in senescent DFCs. Moreover, it is very likely that metabolic processes also interact with the Senescence-Associated Secretory Phenotype (SASP). Here, pro-inflammatory interleukins, secreted by senescent cells, reduce the differentiation potential and immunomodulatory properties of DFCs. This suggests an involvement of pro- and anti-inflammatory factors in osteogenic differentiation, which will be discussed in the next section.

### 3.6. Pro- and anti-Inflammatory Factors

The dental stem cells are said to have immune-modulating properties, which, in addition to their regenerative potential, can also be used for immunotherapies in the future [123]. However, probably in contrast to other dental stem cell types, DFCs are able to both support and regulate the development of osteoclasts during tooth eruption [15]. They can furthermore be influenced by pro-inflammatory components of pathogenic bacteria such as lipopolysaccharide (LPS), which is the major component of the outer membrane of Gram-negative bacteria such as *Escherichia coli*. LPS from *E. coli* did not only induce the expression of pro-inflammatory cytokines such as interleukin 6 but also the activity of alkaline phosphatase, which is an early marker of the osteogenic differentiation [124]. In contrast, LPS from *P. gingivalis* did not induce the expression of proinflammatory cytokines. However, do DFCs have strong immunomodulatory properties and can they also target macrophages in this direction? And how do these properties relate to the osteogenic differentiation potential of DFCs? It is obvious that the same cells of the dental follicle cannot contribute to the bone formation during the formation of the periodontium and bone resorption during tooth eruption at the same time. However, something is known about the immunomodulatory properties of DFCs. Chen et al. showed that DFCs reprogrammed macrophages into the anti-inflammatory M2 phenotype by secreting the paracrine factors TGF-β3 and TSP-1, alleviating LPS-induced inflammation [125]. In addition, DFCs influence also the adaptive immune system. Genç et al. demonstrated that DFCs support an antiproliferative response on CD4^+^ T lymphocytes derived from asthma patients by increasing the amount of CD4^+^CD25^+^FoxP3^+^ regulatory T cells. They conclude that DFCs suppressed allergen-induced polarization of T-helper 2 cells while promoting differentiation of T lymphocytes into T-helper 1 cells [126,127]. Interestingly, the same group obtained similar immunomodulatory effects with human DFCs in a rat sepsis model. Here, treatment with DFCs down-regulated inflammatory responses by lowering TNF-α levels and increasing the number of regulatory T cells [128]. A recent study suggests that especially secreted factors admit the immunomodulatory properties of DFCs since it was sufficient in this study to use DFCs conditioned cell culture media to rescue the regenerative capacity of inflamed rat dental pulp. The conditioned medium down-regulated ERK1/2 and NF-κB signaling pathways in LPS-treated rat dental pulp cells, resulting in suppression of pro-inflammatory IL-1β, IL-6, and TNF-α expression and promotion of IL-4 and TGF-β expression, which are associated with a more immunomodulatory response [129]. How these immune system modulating properties of DFCs relate to their properties as osteogenic progenitor cells remain elusive, but their true dual properties as regulators of tooth eruption and as tissue cells that build tooth roots make DFCs ideal candidates for this endeavor.

## 4. Conclusions

DFCs are dental mesodermal cells, which are involved in both bone destruction during tooth eruption and bone formation during periodontal tissue development. These cells are interesting for regenerative dentistry and a number of works have started to elucidate how DFCs promote osteoclast maturation or osteogenic differentiation. This article summarized works that discovered molecular mechanisms in DFCs. Most of these articles concentrate on mechanisms of osteogenic differentiation. Here, different signaling pathways, epigenetic modifications and ncRNAs could be elucidated. However, these numerous works on specific molecular mechanisms, which are sometimes contradictory, cannot explain the entire process of osteogenic differentiation. For that reason, recent works also focus on crucial biological processes such as reaction to oxidative stress, cell viability, metabolic processes and cellular senescence. This later work is of great interest for the future because it helps us to better understand the whole process of osteogenic differentiation, which may give us some fruitful clues on how, for example, to manipulate biological processes to enhance alveolar bone formation. However, scientific work should also consider the properties of the developing organ, i.e., the developing periodontal ligament or alveolar bone, as a whole. For example, it may be important to consider conflicting effects of soft tissue formation or tissue mineralization on molecular processes in DFCs at the same time. To make more progress, scientists need to think for themselves and not just feed a computer big data.

## Figures and Tables

**Figure 1 ijms-23-05945-f001:**
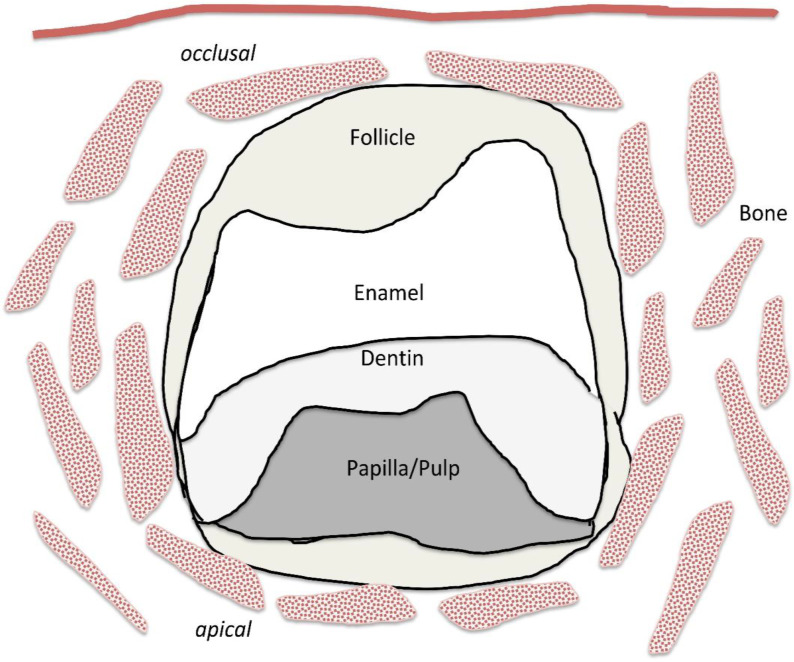
Sketch of an early tooth germ with dental follicle, dental papilla/pulp and already developed mineralized tooth tissue (enamel, dentin) before tooth eruption and tooth root development.

**Figure 2 ijms-23-05945-f002:**
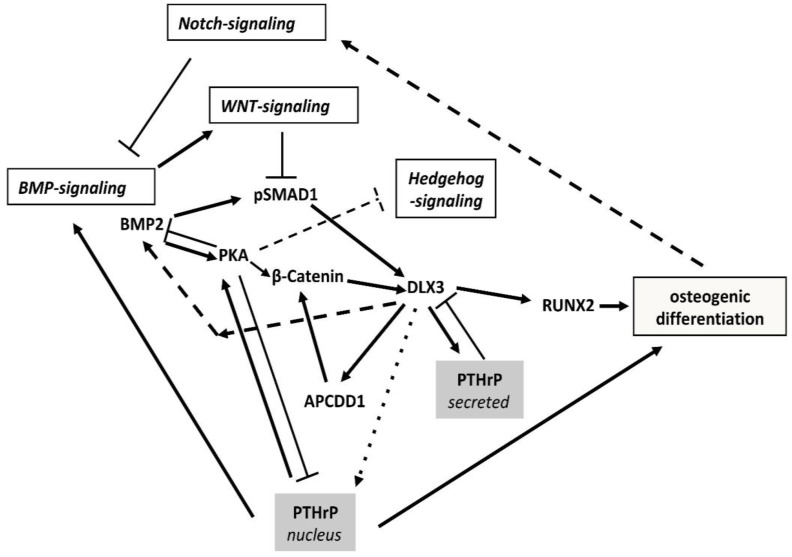
Summary of the BMP2/DLX3 pathway during the osteogenic differentiation of DFCs.

**Figure 3 ijms-23-05945-f003:**
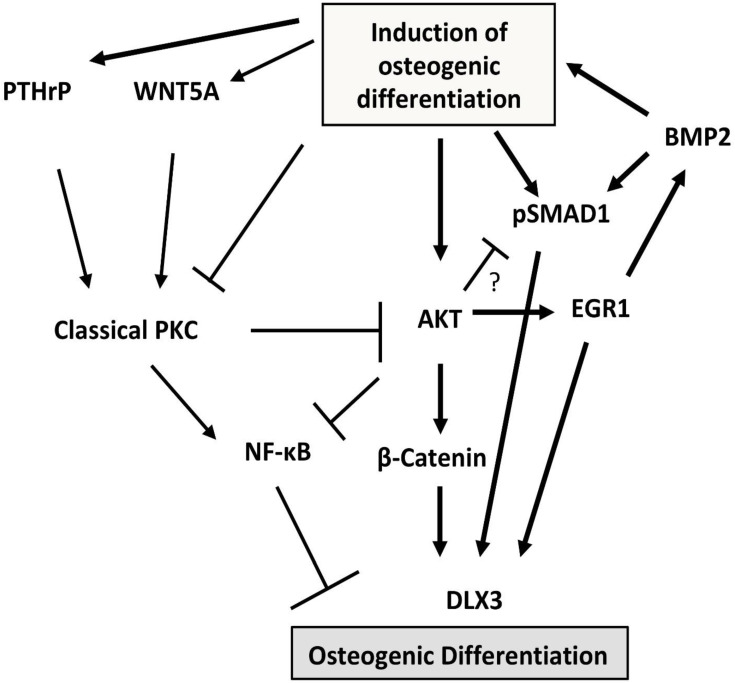
Summary of the PKC/AKT pathway during the osteogenic differentiation of DFC.

## Data Availability

Not applicable.

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
