# Peer review of "Mechanisms during Osteogenic Differentiation in Human Dental Follicle Cells"

_ijms, 2022, doi:10.3390/ijms23115945_

Round 1
Reviewer 1 Report
The abstract is some how very vague; also last sentence said recent studies -but it is not clear what does means here recent. Please re write the abstract
The introduction should be presented in a better structured way in order to indicate the need of actual review;
Please describe how this work is structured in last line of introduction
There is a lot of information but difficult to follow as no structure were presented
Conclusion – first part are little bit confused. For example you said “but it is unclear how DFCs promote” and then you give a potential solution in the following sentence ..
You said recent study in abstract but you describe in references “Friedenstein, A., and Kuralesova, A.I. (1971)”
Author Response
- Reviewer 1 wrote: „The abstract is some how very vague; also last sentence said recent studies -but it is not clear what does means here recent. Please re write the abstract“
Answer: The abstract was re-written with more information about the content of this article.
- Reviewer 1 wrote: „The introduction should be presented in a better structured way in order to indicate the need of actual review“
Answer: Sentences have been included in the introduction to help readers follow the text.
- Reviewer 1 wrote: „Please describe how this work is structured in last line of introduction“
Answer: Added a sentence at the end of the introduction to the general structure of this article. An outline is also added at the beginning of the chapter "Molecular Mechanisms of Osteogenic Differentiation of DFCs".
- Reviewer 1 wrote: „There is a lot of information but difficult to follow as no structure were presented“
Answer: see answer for point 3: Many works have been published in recent years, which is why it is not possible to explain all the studies mentioned in this review article in detail. As a result, the information density is somewhat greater, but this is not unusual for a review article and I think the new inserted structure can give the reader an orientation. In addition, I think I always wrote sentences at the end of each chapter for the transition to the next chapter.
- Reviewer 1 wrote: „Conclusion – first part are little bit confused. For example you said “but it is unclear how DFCs promote” and then you give a potential solution in the following sentence ..“
Answer: Conclusion was revised and I hope things are clearer now.
- Reviewer 1 wrote: „You said recent study in abstract but you describe in references “Friedenstein, A., and Kuralesova, A.I. (1971)”“
Answer: You cannot ignore important studies, even if they are than 50 years old. However, 50 % of all listed references are 5 years or younger.
Reviewer 2 Report
In the present review “Mechanisms during osteogenic differentiation in human dental follicle cells”, Christian Morsczeck summarizes the molecular mechanism of differentiation into mineralizing cells based on recent studies with Human dental follicular cells (DFC). Indeed, he concluded that, these findings are of great interest for the future, because they help us to comprehend better the entire process of the osteogenic differentiation that may give us some fruitful hints how to manipulate biological processes to improve for example the formation of alveolar bone. Moreover, DFCs could be used for studies and research in regenerative medicine and not only in dentistry.
Overall, I think that the manuscript is intriguing, well-written (within the scope of this journal), well-structured and the data are of clinical relevance on a current topic of interest. I would like to congratulate the author on its work.
Author Response
Reviewer2 wrote:“In the present review “Mechanisms during osteogenic differentiation in human dental follicle cells”, Christian Morsczeck summarizes the molecular mechanism of differentiation into mineralizing cells based on recent studies with Human dental follicular cells (DFC). Indeed, he concluded that, these findings are of great interest for the future, because they help us to comprehend better the entire process of the osteogenic differentiation that may give us some fruitful hints how to manipulate biological processes to improve for example the formation of alveolar bone. Moreover, DFCs could be used for studies and research in regenerative medicine and not only in dentistry.
Overall, I think that the manuscript is intriguing, well-written (within the scope of this journal), well-structured and the data are of clinical relevance on a current topic of interest. I would like to congratulate the author on its work.“
Answer to Reviewer2: Thank you for your kind evaluation of my article! I am very honored!
Round 2
Reviewer 1 Report
.